# Satellite Image-Based Methods of Spatiotemporal Analysis on Sustainable Urban Land Use Change and the Driving Factors: A Case Study in Caofeidian and the Suburbs, China

**Guang Yang [1], Sara Chao [1], Jin Yeu Tsou [1] and Yuanzhi Zhang [2,3,\*]**

[1]  Center for Housing Innovations, Chinese University of Hong Kong, Shatin, New Territories,
    Hong Kong, China; 1155106549@link.cuhk.edu.hk (G.Y.); sarachao@cuhk.edu.hk (S.C.);
    jinyeutsou@cuhk.edu.hk (J.Y.T.)
[2]  Key Lab of Lunar Science and Deep-space Exploration, National Astronomical Observatories, Chinese
    Academy of Sciences, Beijing 100101, China
[3]  School of Astronomy and Space Science, University of Chinese Academy of Sciences, Beijing 100049, China
\*  Correspondence: zhangyz@nao.cas.cn; Tel.: +86-10-6480-7833

**Abstract:** As a typical rapid-development seaport area in coastal cities, such as Caofeidian, the study on the spatiotemporal changes of urban land use and its surrounding rural areas is valuable and significant in reference to the future urban planning and land policies in similar coastal areas of China or other countries. Based on satellite images, this research processes images in different years for summarizing the changes of vegetation, urban areas, and water areas in Caofeidian and the suburbs. This research aims to summarize the experience of the coastal city in the process of sustainable development by analyzing the dynamic trends and driving factors of land use spatial and temporal changes in the target area so that it provides a reference for the long-term development of the city. Meanwhile, it also hopes to give support for refining and improving the spatiotemporal analysis method for sustainable urban land use through the experiment. Due to the appearance of the results of the abnormal data, in the experiment process, this article adopts a comparative experiment to avoid the error of the analysis result and to find out the reason. The results show that the urban area for construction increased rapidly in the past twenty years, which is mainly affected by factors, such as economic development, policy guidance, environmental awareness, and environmental protection measures, especially guided by policies. Thus, coastal cities should stretch the planning of sustainable development from the three aspects combining with local characteristics. Besides, phenological phenomena and crops harvest time tremendously affect the images and calculation. The selection of remotely-sensed images should fully consider the characteristics of urban and rural locations, especially the impact of local phenological phenomena. The results of the analysis provide reference value and support for sustainable urban land management and development in the study area and other coastal cities.

**Keywords:** urban land use change; driving factors; phenological phenomena; satellite images; Caofeidian

---

## 1. Introduction

Land resources are the basic space resources for human-being survival and development, and they are inextricably linked with human production activities and economic activities [1]. Meanwhile, changes in landscape patterns caused by land use changes reflect the status and tendency of human

activities [2]. With the economic development and population growth, the demands for construction land are on the rise. It is an essential step in coordinating urban development and rational utilization of land resources to analyze the driving factors of land use changes in an area [3]. Through the dynamic landscape changes reflected by satellite images, policymakers, land planners, and investors or others can make decisions on further actions according to the development characteristics and trends of the region [4–7]. From the perspectives of multidisciplinary and nature, it will also support the protection of coastal urban environment and biodiversity, and the impacts of the control and reduction of the disasters on the local area. This is in line with the concept of green infrastructure and the advocacy of sustainable land management and development [8–11].

Since the contradiction between the land supply and demand has evidently shown up, sea reclamation in the coastal areas has become one of the major strategies for relieving tensions in land use [12]. The land use structure and spatial distribution of Tangshan Caofeidian New District, which locates in Bohai Economic Rim, are widely different from what it was before following by population increase and urban expansion. Coastal areas in many countries, such as Japan, Singapore, the Netherlands, and the United States, have expanded land space by sea reclamation for alleviating conflicts between human need and land supply [13,14]. China has experienced serious ecological and environmental problems after undergoing large-scale reclamation. It can be seen that scientific planning and management have an important impact on urban social and economic development and environmental protection [15]. As a seaport area under the guidance of national policies, the analysis of land use changes in Caofeidian and the suburbs is of great significance for future land planning.

Throughout previous research, the analysis on land use changes in Caofeidian and its suburbs is quite scarce both on breadth and depth. Literature articles written in various fields on Caofeidian land use analysis rapidly increased since the 11th Five-year Plan was issued, after when Caofeidian achieved an excellent development. However, there are only a few hundred academic articles in total and have a downward trend in recent years. In terms of research content, studies based on remote sensing technology in the region involve research on land use changes from multiple perspectives [16], the ecological security assessment based on landscape patterns [17], monitoring and analyzing tidal zone [18], wetland, and coastline changes [19,20]. But there are still very limited studies on the analysis of driving forces in Caofeidian and the suburbs, as the previous studies focus more on the results of land use change. On the other hand, the timeliness of data and the pertinence of study area selection are easily overlooked, which results in a gap of research direction and a lack of reference significance.

Therefore, this article stretches the first analysis on spatial and temporal changes in Caofeidian and its surrounding hinterland in the past 20 years based on remote satellite imagery, which is the important means for the acquisition of accurate land information and spatiotemporal analysis of land resources. It discusses the law of urban land use change, learns the dynamic trends of urban development, investigates the driving factors of land use change, and simultaneously studies the land use changes in the surrounding rural areas in order to discover the mutual effects. The article fully considers the timeliness of data selection via multiple channels to access literature, documents, news reports, and exchanges with local residents to understand the characteristics of the study area, leading to more accurate analysis results of the driving forces and narrowing the gaps.

Besides, it initially uses contrast experiment and analysis to explain the experimental results from a scientific perspective, hoping to provide a reference for subsequent research in other coastal cities of China or other countries. This article would prefer to be complementary in studying spatiotemporal changes in the study area with the perspective of driving factors and altering results, while it would prefer to be considered comprehensively when analyzing by remotely-sensed images.

## 2. Material and Methods

### 2.1. Study Area

The analysis of land use change in this project mainly focused on Caofeidian and its suburbs in the south of Tangshan, Hebei Province, China (Figure 1). The range of study area in the remote sensing experiment was selected for 117°54′25″ E to 118°47′41″ E and 38°52′52″ N to 39°32′23″ N in the south of Tangshan, covering Caofeidian and some areas of Fengnan District, Luannan and Laoting District, as well as part of the sea. Situated in the warm temperate semi-humid continental monsoon climate zone, it has four distinct seasons with cold winter and hot summer. The annual average temperature is 12.7 °C, the average temperature in summer is 25 °C and −0.6 °C in winter. The annual rainfall is 547.2 mm, and the annual average wind speed is 2.4 m/s. The annual 2720.9-h sunshine duration provides excellent light resources for the growth of crops [21]. The harvest time is in June and September each year. The phenological phenomenon is common in the mid-latitude monsoon climate zone. Similar to other cities in the northern part of China, Tangshan has sparse and withered vegetation in winter with the large exposed surface, while lush vegetation and moist soil appear in summer.

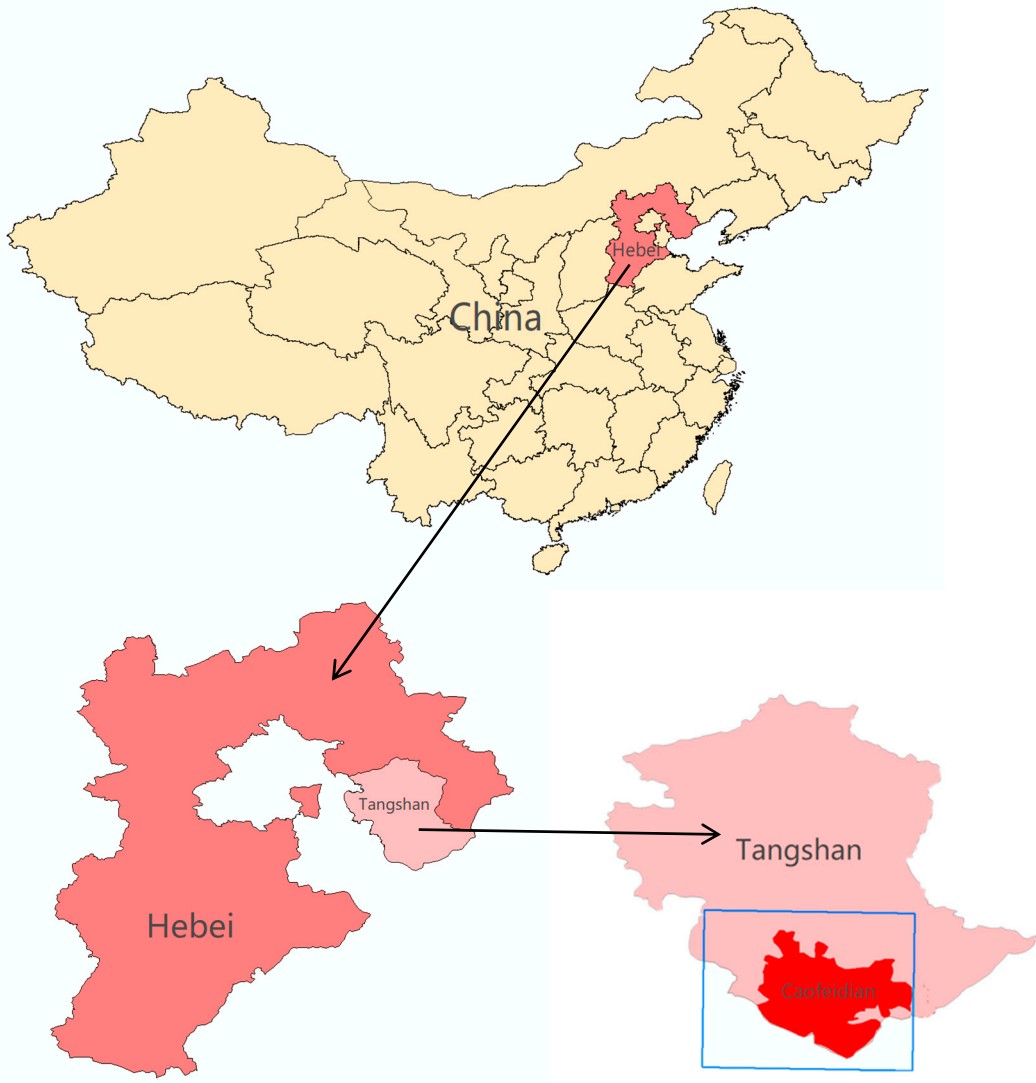

**Figure 1.** Location of the study area (blue border: 117°54′25″ E to 118°47′41″ E and 38°52′52″ N to 39°32′23″ N) and Caofeidian (Source: Author).

The main study area in this article is Caofeidian (Figure 1), located in Bohai Bay, which is a coastal city with a range of latitude and longitude approximately 118°12′12″–118°43′16″ E and 39°07′43″–39°27′23″ N. This area was originally the inlet of the Luan River and formed sand islands after a long period. Under the influence of geographical conditions, such as topography, terrain, soil, and climate, large areas of wetlands, grasslands, and salt fields are distributed in Caofeidian [22]. The region has a high level of salinization since it is close to the sea, which causes huge differences in land use patterns from the north to the south. Many farms are well planted in the northern part of the area, industrial land is distributed in the south, and the construction land for residential use is scattered in Caofeidian. With the adjustment of the administrative division policy and the development in the Caofeidian, the jurisdictional boundaries between counties have changed in the past 20 years [23]. The close cooperation between the three cities Beijing, Tianjin, and Tangshan provides sufficient human and material support for the development and construction of Caofeidian. Under the guidance of national policies, Caofeidian has become one of the regions with rapid urbanization among northern port cities.

*2.2. Urban Planning and Economic Development*

2.2.1. Planning Process

The best natural port in the region of Bohai Bay is located in Caofeidian, which is thought to be a "good location facing the sea with deep troughs and backing on the land with shallows" [24]. As Shougang was about to move into Tangshan, Hebei Provincial government decided to make full use of this opportunity to develop Caofeidian and listed the plan as "No. 1 Project in Hebei Province." in 2003 [25]. In February 2005, the National Development and Reform Commission formally adopted the Shougang relocation plan and agreed to adjust the structure of production in Shougang to control environmental pollution. The steel mill was moved to Tangshan Caofeidian aiming at building a brand new Shougang. In 2008, the Hebei Provincial government approved the construction of the Caofeidian new district. It became a national circular-economy demonstration area with a planned construction area of 380 square kilometers. Then, the State Council approved the cancellation of Tanghai County and established the Caofeidian District in July 2012 [26].

2.2.2. Social-Economic Data

Since Caofeidian was listed as a key development project, remarkable progress has been made in economic and social development in the region, and the comprehensive economic strength has been continuously enhanced. As Caofeidian is the newly established administrative district, the socio-economic data of the region refer to the data of Tanghai County, which is the original administrative district of the main part of Caofeidian. The sources of economic data from 2006 to 2011 were derived from the Tanghai Statistical Yearbook [27], and the data from 2012 to 2016 were derived from the Caofeidian Statistical Yearbook. The total population of Tanghai County grew from 138,271 to 156,907 from 2006 to 2011 with an increase from 748.25 million yuan to 1583.56 million yuan of industrial enterprises above designated size output. The urbanization rate from 2012 to 2016 dramatically rose from 25.96% to 73.28%, indicating the rapid development of urbanization in Caofeidian.

As for the industrial structure, the ratio of the primary industry and the secondary industry among the three industries had changed significantly during the period from 2006 to 2011. The primary industries output accounting for 30.97% in 2006 declined to 18.38% in 2011. On the contrary, secondary industries output increased from 34.63% in 2006 to 41.97% in 2011. However, there was a sharp fluctuation in the output of industrial enterprises above designated size, which displayed a trough and then suddenly increased from 2012 to 2016.

*2.3. Data Collection*

Remotely sensed imagery is the most commonly used source of geographic information data, which can provide a variety of surface feature information at different spatial and temporal scales

through remote sensing technology. In this study, remote sensing data is derived from the USGS (United States Geological Survey) website. Because the study area is in the north of China, the date range is set from August to September when there are less cloud and rains and the vegetation cover is relatively complete. Search Criteria was narrowed to path 122 and row 33 according to the geographical location of the study area in 5 different years (Table 1). The remote sensing data in all the 5 years are Landsat data with the spatial resolution of 30 X 30 m.

**Table 1.** Remote sensing data selected in the project (Source: United States Geological Survey (USGS)).

| Time | Remote Sensing Data | Spatial Resolution (m) | Cloud Cover |
|---|---|---|---|
| **1998-09-01** | Landsat 5 | $30 \times 30$ | 0 |
| **2005-09-04** | Landsat 5 | $30 \times 30$ | 5 |
| **2009-08-30** | Landsat 5 | $30 \times 30$ | 0 |
| **2013-08-25** | Landsat 8 | $30 \times 30$ | 0.72 |
| **2017-09-21** | Landsat 8 | $30 \times 30$ | 4.37 |

The selected remote sensing image in this project complies with the selection principles and foundation of scientific experiments and analysis [28]. It satisfies the requirements of a clear picture, and the cloud coverage is less than 5%. Although the cloud cover in 2005 remote sensing image and in 2017 remote sensing image was as high as 5 and 4.37, respectively, the clouds have not covered and influenced the study area exactly. In addition, the dates of remote sensing images in different years range within one month, which ensures the comparability of images in those five years.

*2.4. Data Preprocessing*

Preprocessing is a significant step in the processing of remote sensing image data. Remote sensing images are distorted due to various factors, such as the variation of satellite speed and the interaction between electromagnetic waves and the atmosphere when acquiring information. Preprocessing of remote sensing image contributes to adjust the blur and distortion in the original image to get the closest results to the real image.

2.4.1. Radiometric Calibration

Radiation calibration is the process of converting grayscale (or Digital Number, abbreviating as "DN") value of the image into physical quantity when users compare the images acquired by different sensors at different times or when calculating the spectral reflectance and the spectral radiance of the surface feature [29]. The software ENVI 5.1 is used as the tool in preprocessing in this project to convert the DN value of the Thematic mapper (TM) images to radiance. The formula is "L (radiance) = gain × DN + bias" [30]. Gain and bias can be read in the header file, which is provided together with the raw data.

2.4.2. Atmospheric Correction

Since the scattering and absorption by atmospheric molecules, aerosols, and cloud particles have an impact on the sunlight transmission process, the information obtained by the sensors contains certain information on non-target surface features. Therefore, removing atmospheric influences is one of the key remote sensing image preprocessing steps. Atmospheric correction aims at eliminating or reducing the influence of atmosphere, light, radiation, etc., on the reflectance and radiation of surface features which will be received by sensors as well as acquiring real physical model parameters. The FLAASH module in ENVI is the preferred atmospheric correction model for reflecting the hyperspectral radiant energy image reflectance. It is applied in the project to calibrate TM images, which shows evident changes in the images (Figure 2). The image after the atmospheric correction has solved the problem of ambiguity of remote sensing images, that is, the spectrum reflects unreal reflectivity of

surface features, and improved the clarity of images, which provides a reliable guarantee for further land cover classification.

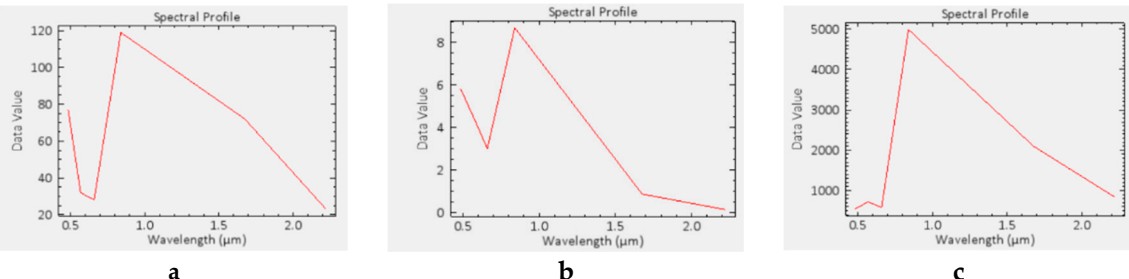

**Figure 2.** Vegetation spectral of the original image (**a**), Calibrated image (**b**), Atmospheric correction image (**c**) (Source: Author).

### 2.4.3. Image Clip

The area of Caofeidian district has continuously changed, and the administrative divisions are in a state of constant adjustment, so this project chose a study area covering Caofeidian and the suburbs by regular clipping using the ENVI 5.1. The vector data is set to a range of 117°54′25″ E to 118°47′41″ E and 38°52′52″ N to 39°32′23″ N. The final remote sensing image after the operation is shown below (Figure 3). The red line shows the general boundary of the current administrative division in Caofeidian for reference.

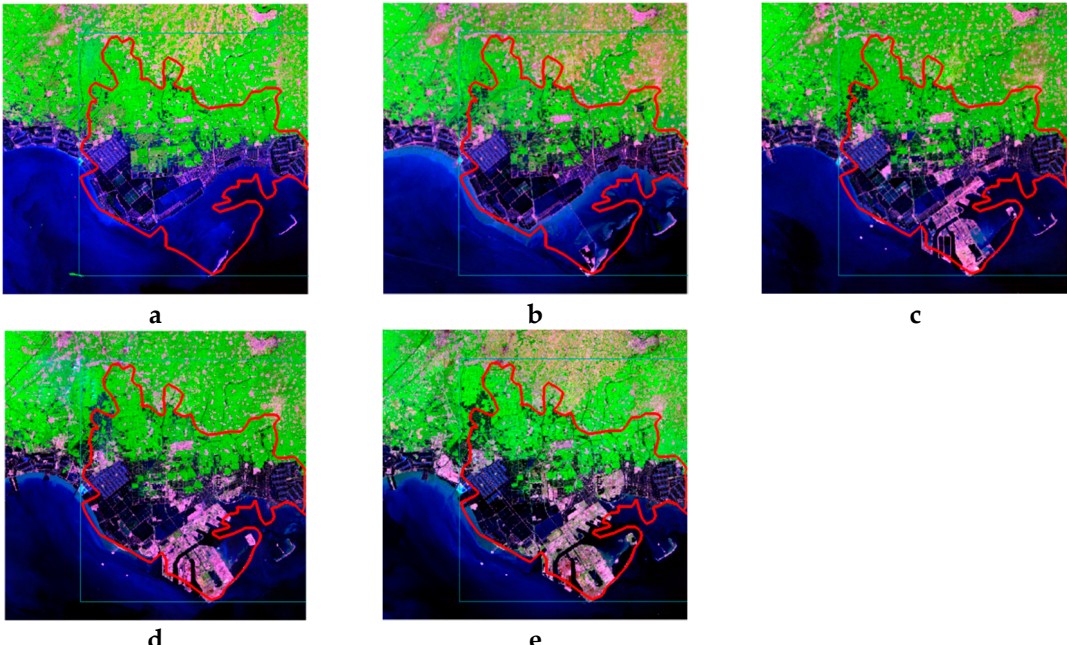

**Figure 3.** Remote sensing image (117°54′25″ E to 118°47′41″ E and 38°52′52″ N to 39°32′23″ N) (Source: United States Geological Survey (USGS)). (**a**) 1998 RGB: 7:4:3, (**b**) 2005 RGB: 7:4:3, (**c**) 2009 RGB: 7:4:3, (**d**) 2013 RGB: SWIR2: NIR: Red, (**e**) 2017 RGB: SWIR2: NIR: Red.

## *2.5. Image Classification*

### 2.5.1. Land Use Patterns Definition

Establishing a reasonable and scientific classification system for land use and land cover is the premise for carrying out research on land use change. The classification system requires the comprehensive consideration of the identifiability of remote sensing data and the characteristics of

regional land use, which are even the purpose of research. The representative classification system at home and abroad are existing for reference, but the classification system in each project also needs to be built according to the actual situation.

The characteristics of diverse land use in the study area are a large part of cultivated land and grassland and a few woodland, the large field covered by the river surface, reservoir area, and aquaculture area, and the continuous increase in the reclaimed land [31] (Table 2). Based on its characteristics and the demands of this project, the land use patterns are merged into three types—vegetation, urban area, and water referring to the Current land use classification GB/T21010-2017.

**Table 2.** Land use patterns' definition (Source: Author).

| Land Use Pattern | Description |
| --- | --- |
| Vegetation | Cultivated land, grassland, woodland, wetlands |
| Urban | Construction land, impervious surface, reclaimed land, bare land |
| Water | River, sea, reservoir, aquaculture area, salt field, tidal flat |

### 2.5.2. Support Vector Machine (SVM) Classification

Support vector machine is a new method type of data mining, which is widely used in pattern recognition and machine learning [32,33]. In recent years, SVM classification has excellent performance in image classification, which improves generalization ability and solves high-dimensional problems and nonlinear problems. SVM classification is used in this project.

### *2.6. Accuracy Assessment Method*

Accuracy assessment refers to comparing field data with classification results to confirm the accuracy of the classification process [34]. Accuracy assessment of classification results is a critical part of the remote sensing monitoring of land use, and it is also a measure of whether the classification result is credible.

### 2.6.1. Confusion Matrix

A confusion matrix is an N × N matrix that is used to analyze the error of interpretation results by comparing the reference point and the classification point. Generally, rows represent the classification points, and the columns represent the reference points. It is impractical to detect every pixel in a classified image. Thus, it is necessary to select a set of reference pixels randomly. The test of the interpretation accuracy in this article was mainly calculated by ENVI 5.1. It contrasts the random selection of pixels referring to the land use atlas at the same period and Baidu map image to classification results. The confusion matrix in this project is shown below (Table 3).

**Table 3.** Confusion matrix (Source: Author).

| Classification | Ground Truth ROI | | | |
| --- | --- | --- | --- | --- |
| | Region1 (Vegetation) | Region2 (Urban) | Region3 (Water) | Total |
| Vegetation | VR1 | VR2 | VR3 | V |
| Urban | UR1 | UR2 | UR3 | U |
| Water | WR1 | WR2 | WR3 | W |
| Total | R1 | R2 | R3 | |

2.6.2. Overall Accuracy (OA)

Overall Accuracy (OA) represents the probability that the classification result of each random sample corresponds to the actual surface. Its result equals to the sum of correctly classified cells divided by the total number of pixels. The number of correctly classified pixels is distributed along the diagonal of the confusion matrix. VVR1, UR2, and WR3 (Table 3) are used for overall accuracy calculation. The formula is:

$$OA = \frac{VVR1 + UR2 + WR3}{V + U + W} * 100$$

2.6.3. Kappa Coefficient

Unlike the evaluation of the error of the main diagonal elements by overall accuracy, Kappa coefficient refers to an index that measures the degree of coincidence or accuracy between two classified images and comprehensively measures the classification error [35]. The result of the Kappa coefficient calculation can be divided into different levels indicating the consistency. For instance, the number from 0.81 to 1.00 manifests the almost perfect accuracy of the classification result.

## 3. Results and Analysis

After preprocessing and classification processing of remote sensing images, the results were calculated through ENVI 5.1 and ArcGIS 10.2 for analysis. Although the process of the experiment was correct and the data were error-free, an abnormal phenomenon was still found in this project. In this regard, the study also added a comparative experiment for further analysis to find out the causes and made a reasonable explanation.

### 3.1. Land Use Pattern Classification Result

Followed by the steps of processing in ENVI 5.1, the results of classification are converted to tiff format suitable for ArcGIS and achieving image results visualization. Vegetation area is set in the color of green, the urban area is set in red, and water is in blue (Figure 4). From the data results of the overall accuracy and Kappa coefficient, the classification accuracy of remote sensing images over the five years is relatively high, that is, all of them are over 99% and 0.99 meeting the requirements of this study (Table 4). On the other hand, it also indirectly shows the high efficiency and validity of the classification method.

**Table 4.** Accuracy assessment result (Source: Author).

|  | 1998 | 2005 | 2009 | 2013 | 2017 |
|---|---|---|---|---|---|
| OA (Overall Accuracy) | 99.6205% | 99.5854% | 99.7166% | 99.6678% | 99.4065% |
| Kappa | 0.9937 | 0.9936 | 0.9957 | 0.9947 | 0.9910 |

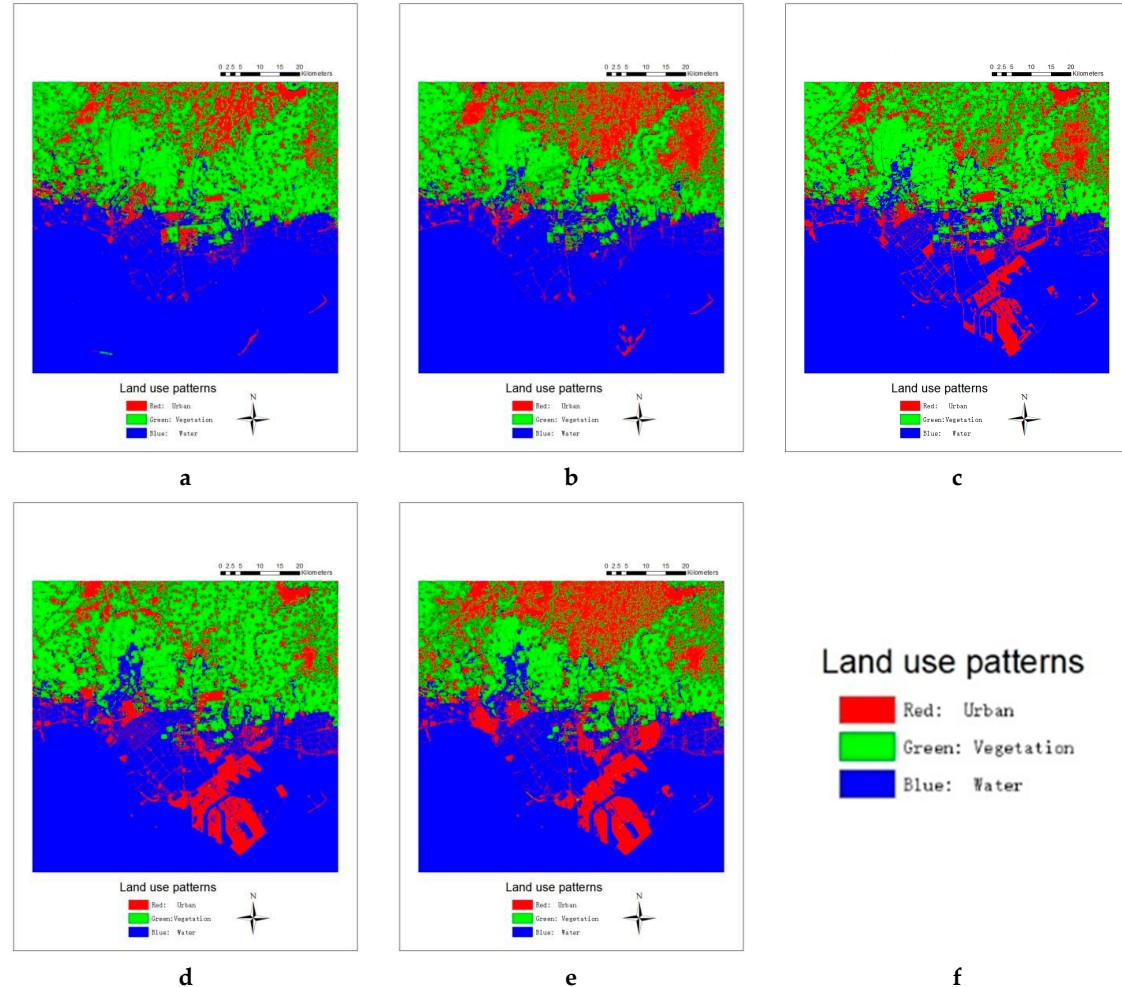

**Figure 4.** Land cover maps of 1998, 2005, 2009, 2013, and 2017 (117°54′25″ E to 118°47′41″ E and 38°52′52″ N to 39°32′23″ N) (Source: Author). (**a**) 1998 Land Cover around Caofeidian, (**b**) 2005 Land Cover around Caofeidian, (**c**) 2009 Land Cover around Caofeidian, (**d**) 2003 Land Cover around Caofeidian, (**e**) 2017 Land Cover around Caofeidian, (**f**) Legend of Land use patterns.

## 3.2. Land Use Statistic and Land Cover Transformation

### 3.2.1. Calculation of Areas

After the calculation by ENVI 5.1, the results on the percentage of vegetation, urban, and water in 1998, 2005, 2009, 2013, and 2017 are shown below (Tables 5–9). However, the data results also display an abnormal trend of land use changes. From 1998 to 2005, the proportion of vegetation area fell from 30.74% to 26.157%, but the proportion of the area increased to 30.301% from 2005 to 2009. Similarly, the proportion of vegetation area from 2013 to 2017 quickly dropped from 31.593% to 25.13%. For such changes and results, the actual land use situation and changes in the local area are not considered to be accurate.

**Table 5.** Land cover statistics in 1998 (Source: Author).

| 1998 | Vegetation | Urban | Water |
|---|---|---|---|
| Points | 1,919,914 | 928,314 | 3,395,966 |
| Percentage | 30.74% | 14.867% | 54.386% |
| Area (m$^2$) | 1,727,922,600 | 835,482,600 | 3,056,369,400 |

**Table 6.** Land cover statistics in 2005 (Source: Author).

| 2005 | Vegetation | Urban | Water |
|---|---|---|---|
| Points | 1,633,281 | 1,142,923 | 3,467,900 |
| Percentage | 26.157% | 18.304% | 55.539% |
| Area(m$^2$) | 1,469,952,900 | 1,028,630,700 | 3,121,191,000 |

**Table 7.** Land cover statistics in 2009 (Source: Author).

| 2009 | Vegetation | Urban | Water |
|---|---|---|---|
| Points | 1,892,035 | 1,152,899 | 3,199,260 |
| Percentage | 30.301% | 18.464% | 51.236% |
| Area(m$^2$) | 1,702,831,500 | 1,037,609,100 | 2,879,334,000 |

**Table 8.** Land cover statistics in 2013 (Source: Author).

| 2013 | Vegetation | Urban | Water |
|---|---|---|---|
| Points | 1,972,744 | 1,124,696 | 3,146,754 |
| Percentage | 31.593% | 18.012% | 50.395% |
| Area(m$^2$) | 1,775,469,600 | 1,012,226,400 | 2,832,078,600 |

**Table 9.** Land cover statistics in 2017 (Source: Author).

| 2017 | Vegetation | Urban | Water |
|---|---|---|---|
| Points | 1,569,162 | 1,579,054 | 3,095,978 |
| Percentage | 25.130% | 25.288% | 49.582% |
| Area(m$^2$) | 1,412,245,800 | 1,421,148,600 | 2,786,380,200 |

### 3.2.2. Contrast Experiment

Drastic changes are extremely unusual in land use in only four years. By investigating the local area and getting information from local people and the news from Tangshan Bureau of Agriculture & Animal Husbandry, it was found that the period from the end of August to the beginning of September is the harvest time of crops. Additionally, since there are many farms in the study area, large-scale mechanized operations are quite convenient and efficient. According to the records from the official website, it only calls for five to seven days to complete the wheat harvest and about 20 days to complete the peanut seeding or harvesting under the mechanized management [36].

The remote sensing images in 2009 and 2013 were obtained on August 30, 2009 and August 25, 2013, respectively, when the crop might not have been harvested. Therefore, the remote sensing image of September 15, 2009 was selected in the comparative experiment. The information of comparative image on September 15, 2009 is the same as the 2009 data in the original experiment. Although the cloud cover of 2009-09-15 remote sensing image is seven, the cloud layer does not cover above the study area, which has no effect on the analysis of land cover change.

The results of 2009-09-15 remote sensing image after data preprocessing, including radiometric calibration, atmospheric correction, image clip, etc., and image classification, accuracy assessment analysis, and other steps are compared with the remote sensing image on August 30, 2009. The details are listed in Table 10.

By comparison, the hypothesis that the crops were not harvested on August 30, 2009 was found to be reasonable.

**Table 10.** Comparison between images of 2009-08-30 and 2009-09-15 (Source: Author).

| Items | 2009-08-30 Land Cover around Caofeidian | 2009-09-15 Land Cover around Caofeidian |
|---|---|---|
| Land cover map |  |  |
| OA | 99.7166% | 99.5285% |
| Kappa | 0.9957 | 0.9929 |
| Points of vegetation | 1,892,035 | 1,489,400 |
| Percentage of vegetation | 30.301% | 24.006% |
| Area of vegetation | 1,702,831,500 | 1,340,496,000 |
| Points of urban | 1,152,899 | 1,536,948 |
| Percentage of urban | 18.464% | 24.772% |
| Area of urban | 1,037,609,100 | 1,383,253,200 |
| Points of water | 3,199,260 | 317,992 |
| Percentage of water | 51.236% | 51.222% |
| Area of water | 2,879,334,000 | 2,860,192,800 |

### 3.2.3. Land Use Transformation

Due to the bias appearing in the research and analysis, the results of land use transformation were selected from the four-year data of 1998, 2005, 2009-09-15, and 2017. The crop harvesting almost had been done before the obtained dates of these remote sensing images.

The classification results of the land cover calculated by ENVI 5.1 was converted to a format that can be opened in ArcGIS 10.2. The data of land use changes were calculated by the tools, such as the Map Algebra Tool. The data were calculated through the algebraic formula. The example is the land use transformation from 1998 to 2005:

1998 to 2005 land use transformation = "1998 land cover" × 10 + "2005 land cover".

The calculation results of land use change from 1998 to 2005, 2005 to 2009-09-15, and 2009-09-15 to 2017 are as follows (Tables 11–13):

**Table 11.** Land use transformation from 1998 to 2005 (Source: Author).

| 1998–2005 | Vegetation | Urban | Water |
|---|---|---|---|
| Vegetation | 1,279,350,000 | 160,003,800 | 30,599,100 |
| Urban | 367,660,800 | 602,289,900 | 58,680,000 |
| Water | 80,911,800 | 73,188,900 | 2,967,090,300 |

**Table 12.** Land use transformation from 2005 to 2009-09-15 (Source: Author).

| 2005–2009-09-15 | Vegetation | Urban | Water |
| --- | --- | --- | --- |
| Vegetation | 1,131,682,500 | 176,451,300 | 32,362,200 |
| Urban | 286,684,200 | 805,859,100 | 290,701,800 |
| Water | 33,246,000 | 26,870,400 | 2,786,325,300 |

**Table 13.** Land use transformation from 2009-09-15 to 2017 (Source: Author).

| 2009-09-15–2017 | Vegetation | Urban | Water |
| --- | --- | --- | --- |
| Vegetation | 1,064,515,500 | 305,896,500 | 23,676,300 |
| Urban | 20,263,500 | 983,393,100 | 214,742,700 |
| Water | 73,350,000 | 93,955,500 | 2,608,022,700 |

In addition, the land use transformation map clearly and intuitively reflects the conversion relationship between land use patterns. In this project, land use changes are divided into five groups—unchanged land, vegetation (converts) to urban, water (converts) to urban, vegetation and urban (convert) to water, and urban and water (convert) to vegetation, convenient for subsequent analysis, discussion, and conclusion (Figure 5).

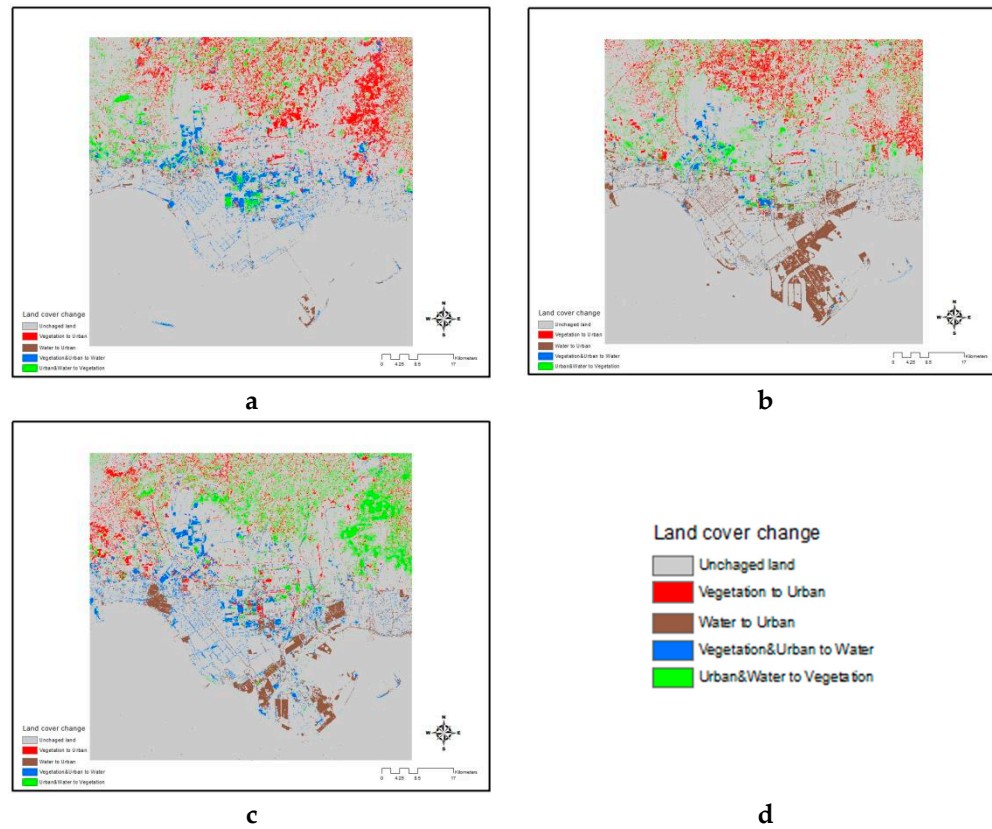

**Figure 5.** Land use transformation maps. (117°54′25″ E to 118°47′41″ E and 38°52′52″ N to 39°32′23″ N) (Source: Author). (**a**) 1998 to 2005 Land use transformation, (**b**) 2005 to 2009-09-15 Land use transformation, (**c**) 2009-09-15 to 2017 Land use transformation, (**d**) Legend of Land cover change.

Seeing from the data and land use transformation map, the changes in land use in Caofeidian and the suburbs in the past 20 years have unique features. From 1998 to 2005, the area of vegetation converted into urban areas reached 367,660,800 m$^2$, and the 286,684,200 m$^2$ conversion appeared in the next phase from 2005 to 2009-09-15. The more important and more prominent feature showed in the second phase is that the urban area converted from the water area was 29,070,800 m$^2$, which is more

than the area converted from vegetation to urban area. The brown part can also be clearly recognized from the map, which refers to the area that is converted from water to urban area. The characteristics of land use transformation in the third phase from 2009-09-15 to 2017 are different from the previous two phases. The area converted by the urban into vegetation is much larger than the area converted by other land use patterns. The converted area was 305,896,500 m$^2$, while the area converted from water to urban areas was still large within 214,742,700 m$^2$.

## 4. Discussion

### *4.1. Driving Factors of Land Use Change*

The factors affecting land use change are complex and varied, by which internal and external conditions will drive the change of land use structure. Overall, the urban area in this study region has been continuously increasing, while the area of vegetation has been gradually decreasing. However, the proportion of vegetation area has been relatively stable from 2009 to 2017. The land use of water area in the study area also showed a decreasing trend in the past two decades.

#### 4.1.1. Driving Factors of Land Use Change in First Stage (1998 to 2005)

From the experimental results, the land use change from 1998 to 2005 was mainly converted from vegetation to urban area. By reference from the social and economic data of Tanghai County, the GDP of the region increased from 2215.35 million yuan to 2852.27 million yuan. The economic growth prompted the expansion of urban and rural areas with the most conversion of farmland into residential areas. At the same time, economic development also promoted the development of local transportation. The attention of the local government towards the construction of rural roads is also the reason for land use changes. As judged by the area and extent of land use change, this stage is a period of slow urbanization. The area of water is less affected by the slow process of urbanization and basically remains in the same state as before, which shows that the urbanization in the region does not involve the development of water area. Economic development is the main driving factor of land use change during this period.

#### 4.1.2. Driving Factors of Land Use Change in Second Stage (2005 to 2009)

From 2005 to 2009, the land use change in the study area was relatively sharp, showing obvious characteristics in urbanization. Land use patterns of the original vegetation and water areas had transformed into construction land. The area where waters are converted into urban even exceeded the area where vegetation is converted into urban. This phenomenon was mainly caused by the statement of relevant policies by the national government in 2005 proposing the goal of "building an iron and steel conglomerate with international advanced level in Caofeidian". On the Labor Day in 2007, Premier Wen Jiabao visited Caofeidian and asked to work hard to build Caofeidian into a "bright pearl" in the Bohai Rim region. The construction of the port in Caofeidian continued to accelerate, and investment in land reclamation had reached more than 2 billion yuan. Policy guidance is the biggest driving factor in the rapid process of urbanization. Just in four years, this area changed from a major agricultural region to a port city with the fast growth of the industry. It can be concluded that changes in the land policy have greatly influenced the transformation of land use patterns in an area, further causing changes in the production and lifestyle of the entire region.

#### 4.1.3. Driving Factors of Land Use Change in the Third Stage (2009 to 2017)

From 2009 to 2017, the conversion area of urban to vegetation increased significantly. Economic development would often bring negative impacts on the environment and challenges to sustainable development. In the past decade, especially after the 2008 Beijing Olympic Games, the country placed more focus on ecological development and environmental protection. Caofeidian wetland is the largest coastal wetland in the north with international significance. In the Caofeidian overall plan from 2008

to 2020, it is stated that the restoration of wetlands and the formation of regional country parks as ecological "green lungs". Regardless of aspects of policy or environment, both the government and the local people are aware of the importance of environmental protection and ecological development. They also try their best for sustainable environmental development. Therefore, new progress has been made in the greening of the research area. On the other hand, the urban planning guidelines for land reclamation keeps going on, but the conversion of water into urban area has apparently slowed down. The process of port construction is mainly influenced by policies and economic development. After 2013, the slower economic and social development in Caofeidian and its suburbs has also been an important reason for the reduction in land use change. However, from a macro perspective, this does not mean that urbanization has regressed but reflecting that the development of urbanization has entered a relatively steady phase.

Through the discussion above, the improvement of urbanization, economic level, and awareness of environmental protection will have an impact on the slow changes in land use, and the policy will dramatically affect the rapid changes in land use patterns. Driven by these factors, Caofeidian formed a very different situation from the original growth in the past 20 years.

### 4.2. Effect of Local Phenological Phenomenon on Image Analysis

As an important part of the phenological phenomenon, plant phenology visually reflects changes in agricultural production and environmental conditions [37,38]. The remote sensing images in the mid-latitude monsoon climatic regions also greatly reflect the regional phenological phenomenon according to the results of the comparative experiment. Since no crop or vegetation-covered land is indistinguishable from urban impervious surfaces in remote sensing image classification, phenological phenomena cause certain deviations from the analysis results [39]. In this project, the experiment selects all the classification results from the images on the first half of September to conduct research to ensure the same situation as far as possible. However, due to the lack of mechanized harvesting or the difference in harvesting time in a small number of areas, the classification of urban construction land and vegetation land will still be affected. Fortunately, the harvest time of crops in the study area is concentrated in a short period of about 20 days from the end of August to the beginning of September. The effects of local phenology caused by human activities on remote sensing images and analysis results can be considered and avoided in the experiment.

## 5. Conclusions

This article analyzes the spatial-temporal changes of urban land use from the satellite images in Caofeidian and suburbs of China. Through this research, it is found that the main driving factors of urban land use change in Caofeidian New District and the suburbs are the urbanization and economic development, policy guidance, and environmental awareness and measures, which are consistent with China national conditions. Those are also important driving forces for land use development in other coastal cities. As far as the current national policies and measures for the coastal cities development are concerned, they pay more attention to environmental protection on the urban sustainable development and planning, which is also an important task of the 13th Five-Year Plan. The results deepen the important role of land use change in reflection and reaction in the development trend of land use pattern. Therefore, Caofeidian and its suburbs and even other coastal cities in China should focus on urban development trends and national policies stretching scientific planning to promote sustainable development.

In addition, it is found that the phenological phenomenon affects the analysis of land cover changes in remotely-sensed images after comparative experiments. The characteristics of the crop growth period, crop planting area, and harvest time in the study area are part of the key influencing factors in the analysis of land use. Therefore, when selecting satellite images, the characteristics and influential factors of the study area should be fully considered. However, it still needs to be refined for

the methods applied to sustainable land management and development of urban land use in other coastal areas in China or other countries.

**Author Contributions:** G.Y. and Y.Z. conceived, designed, and performed the experiments, analyzed the data, and wrote the paper; S.C. improved the data analysis; J.Y.T. contributed reagents/materials/analysis tools.

**Funding:** This research is jointly supported by the National Key Research and Development Program of China (Project Ref. No. 2016YFB0501501), the Pioneer Science and Technology Special Training Program B (No. XDPB11-01-04) of Chinese Academy of Sciences, and the China-Italy Collaborate Project for Lunar Surface Mapping (No. 2016YFE0104400).

**Acknowledgments:** The authors are extremely grateful to the Landsat TM from the US Geological Survey (USGS), the regional map, and the vector map of Tangshan from local government. This research is jointly supported by the National Key Research and Development Program of China (Project Ref. No. 2016YFB0501501), the Pioneer Science and Technology Special Training Program B (No. XDPB11-01-04) of Chinese Academy of Sciences, and the China-Italy Collaborate Project for Lunar Surface Mapping (No. 2016YFE0104400).

**Conflicts of Interest:** The authors declare no conflict of interest.

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
