# Peer review of "Satellite Image-Based Methods of Spatiotemporal Analysis on Sustainable Urban Land Use Change and the Driving Factors: A Case Study in Caofeidian and the Suburbs, China"

_sustainability, doi:10.3390/su11102927_

Round 1
Reviewer 1 Report
The paper discusses a very current issue of spatial and temporal changes in sustainable urban land use. The topic is enduring and critical for local development of coastal regions. Unfortunately, the results of study seem to be of the secondary importance to the authors.
The title of the paper does not reveal the text, as the main emphasis is put on the methods. The results of the research can be only find in the discussion part. I would recommend reading the title carefully and focus on what is important.
The aim of the paper should be clarified, is it to analyse the spatial and temporal changes in land use or it is to discuss methods and show their low relevance and the need to refine.
The paper aims to give the driving factors of changes, isn’t the population growth itself a driving factor ? The reader will easily find the latitude and longitude but there is no information of the social data on the studied city. The Figure 1 gives a very little information to the reader.
In the introduction part similarly to the whole text there is no reference to any other regions and studies. The references include only the study area and methods. It is difficult to say that few hundred academic articles is a small number for one study area. On the other hand there is no information on similar research in the other regions or countries, what is of great significance as authors aim to make their research more general. These results, in my opinion, are not prone to generalization.
As for the chosen time it seems to be short to present changes in land use patterns and it is not well explained why the date range from August to September is adequate of show the role of phenological phenomena.
It is also not clear why the authors decided to use the classification of land use which is not the best in terms of the project. It is already mentioned in the abstract that the distinction between rural bare soil and urban construction area is not so obvious. The reader will find bare land as a urban land in the paper.
In conclusions it can be read that as environmental awareness improved, the vegetation area increased. This should be described better as seems to be very arguable.
There are no sources given under the figures and tables.
Author Response
Dear Reviewer 1:
Thank you very much for your comments. We have revised and updated based on your comments.
Your sincerely,
Authors

Reviewer 2 Report
The manuscript entitled "Analysis on Spatial and Temporal Change in Sustainable Urban Land Use from Satellite Images: A case study in Caofeidian and the Surroundings, China", by G. Yang, S. Chao, J.Y. Tsou & Y. Zhang, presents an interesting work.
In general, the manuscript should be acceptable for publication but some serious problems must be repaired prior to publication. It needs some significant improvement. Some suggestions are as follows:
Please use different terms in the “Title” and the “Keywords”.
I suggest to rewrite the conclusions, In this section, please, write your conclusions with a distinct way.
The English language usage should be checked by a fluent English speaker. It is suggested to the authors to take the assistance of someone with English as mother tongue.
You could enrich the scientific literature. It is so poor.
Please justify convincingly why this manuscript (method, thematology etc) connected with Sustainability’s content and scope. Perhaps the using of proper literature would be helpful. Eg.
- Castanho, R.A.; Naranjo Gómez, J.M.; Kurowska-Pysz, J. Assessing Land Use Changes in Polish Territories: Patterns, Directions and Socioeconomic Impacts on Territorial Management. Sustainability 2019, 11, 1354.
- Skilodimou, H.D.; Bathrellos, G.D.; Koskeridou, E.; Soukis, K.; Rozos, D. Physical and Anthropogenic Factors Related to Landslide Activity in the Northern Peloponnese, Greece. Land 2018, 7, 85.
"Fig. 5 Land use transformation maps". Are you meaning Land use changes
The Headings are Results and Conclusions (plural).
In the title, you are writing "Surroundings". Do you mean "Suburbs"?
It would be useful to be described the aim of this paper.
The authors could make a discussion in the introduction with the hazards - risks which could be caused by the land use changes. See the following publications:
- Bathrellos, G.D.; Skilodimou, H.D.; Chousianitis, K.; Youssef, A.M.; Pradhan, B. Suitability estimation for urban development using multi-hazard assessment map. Sci Total Environ 2017, 575, 119-134.
In all maps you must put coordinates.
Correct breferences in the text and the reference list according to the journal’s format. Please format the references’ list by using the correct journal abbreviations.
See the following link: http://images.webofknowledge.com/WOK46/help/WOS/A_abrvjt.html
Author Response
Dear Reviewer 2:
Thank you for your comments.
According to your comments, we have revised and updated the manuscript as re-submitted.
Attached is the reply to all comments.
Best regards,
Authors

Round 2
Reviewer 1 Report
The paper has been modified according to the reviewers suggestions, however, in my opinion the whole paper still focuses on methods rather than the research results. I would rather say that in the title e.g "Satellite image based methods of spatiotemporal analysis ...."
Author Response
Dear Reviewer 1:
Thank you for your comments. We have revised and improved the title as you suggested.
Attached is the reply to your comments and the revised manuscript.
Yours sincerely,
Authors

Reviewer 2 Report
The manuscript entitled "Analysis on Spatiotemporal Change and the Driving Factors in Sustainable Urban Land Use from Satellite Images: A case study in Caofeidian and the Suburbs, China", by G. Yang, S. Chao, J.Y. Tsou, Y. Zhang, presents an improved and good work.
The manuscript should be acceptable for publication in the present form.
Author Response
Dear Reviewer 2:
Thank you for your comments.
Best regards,
Authors
